# Modification of Lipid-Based Nanoparticles: An Efficient Delivery System for Nucleic Acid-Based Immunotherapy

**DOI:** 10.3390/molecules27061943

**Published:** 2022-03-17

**Authors:** Chi Zhang, Yifan Ma, Jingjing Zhang, Jimmy Chun-Tien Kuo, Zhongkun Zhang, Haotian Xie, Jing Zhu, Tongzheng Liu

**Affiliations:** 1College of Pharmacy, The Ohio State University, Columbus, OH 43210, USA; zhang.9395@buckeyemail.osu.edu (C.Z.); kuo.249@buckeyemail.osu.edu (J.C.-T.K.); zhang.5763@osu.edu (Z.Z.); 2William G. Lowrie Department of Chemical and Biomolecular Engineering, The Ohio State University, Columbus, OH 43210, USA; ma.1711@buckeyemail.osu.edu (Y.M.); zhang.8211@osu.edu (J.Z.); 3Department of Statistics, The Ohio State University, Columbus, OH 43210, USA; xie.908@osu.edu; 4College of Nursing and Health Innovation, The University of Texas Arlington, Arlington, TX 76010, USA; 5College of Pharmacy, Jinan University, Guangzhou 511443, China

**Keywords:** lipid-based nanoparticles, drug delivery, immunotherapy, nucleic acids

## Abstract

Lipid-based nanoparticles (LBNPs) are biocompatible and biodegradable vesicles that are considered to be one of the most efficient drug delivery platforms. Due to the prominent advantages, such as long circulation time, slow drug release, reduced toxicity, high transfection efficiency, and endosomal escape capacity, such synthetic nanoparticles have been widely used for carrying genetic therapeutics, particularly nucleic acids that can be applied in the treatment for various diseases, including congenital diseases, cancers, virus infections, and chronic inflammations. Despite great merits and multiple successful applications, many extracellular and intracellular barriers remain and greatly impair delivery efficacy and therapeutic outcomes. As such, the current state of knowledge and pitfalls regarding the gene delivery and construction of LBNPs will be initially summarized. In order to develop a new generation of LBNPs for improved delivery profiles and therapeutic effects, the modification strategies of LBNPs will be reviewed. On the basis of these developed modifications, the performance of LBNPs as therapeutic nanoplatforms have been greatly improved and extensively applied in immunotherapies, including infectious diseases and cancers. However, the therapeutic applications of LBNPs systems are still limited due to the undesirable endosomal escape, potential aggregation, and the inefficient encapsulation of therapeutics. Herein, we will review and discuss recent advances and remaining challenges in the development of LBNPs for nucleic acid-based immunotherapy.

## 1. Introduction

In recent decades, gene therapy that involves either replacing a mutated gene that functions abnormally or regulating the expression of one or more specific genes has been widely explored for the treatment of various diseases, such as cancers, inherited diseases, infections, and immunodeficiencies [1]. At present, the strategy of gene therapy is to deliver genetic agents, such as antisense oligonucleotides (ASOs), small interfering RNA (siRNA), messenger RNA, micro-RNA (miRNA), and plasmid-DNA, into cells and target the nucleus or cytosol [2,3]. ASOs, as a therapeutic constituent, are generally short, synthetically modified DNAs or RNAs, comprising 8–50 nucleotides in length, that can target mRNA or pre-mRNA through complementary base pairing [4]. ASOs can regulate the expression of target mRNA via RNase H-mediated gene translation reduction, a direct steric hindrance to mRNA, or exon-modulated splicing alteration [4,5,6,7,8,9]. Compared to other small molecules, the advantage of ASOs is that their sequence can be modified to precisely target specific mRNA directly, which can enhance the targeting specificity and is less likely to cause off-target effects or activate the host immune system [10]. With the development of ASO modifications over the years, more druggable targets of ASOs have been identified and, therefore, contribute to potential clinical applications of antisense therapy [8]. 

RNA interference (RNAi) was discovered in 1998 by Andrew Fire and Craig C. Mello, who won the Nobel Prize in physiology or medicine in 2006 for their contribution to the discovery of RNAi [11,12]. In general, RNAi exists in eukaryotes and is mediated by RNA-induced silencing complex (RISC), which is formed from the guide strand of siRNA cleaved by Dicer [3,13,14]. RNAi includes two types of RNA materials, small interfering RNA (siRNA) and microRNA (miRNA). siRNA is a double-stranded non-coding RNA with a length of 20–30 base pairs [15]. It targets mRNA and silences target gene to prevent the subsequent translation process [16]. In the past decade, siRNAs have been widely used for the treatment of various cancers and inherited diseases; the most famous example is patisiran [17]. miRNA is another main type of RNAi molecule, which functions as a gene mediator to regulate specific gene expression in the post-transcription process. The biological structure of miRNAs is similar to siRNAs, and they share the same RNAi downstream pathway. The difference is that miRNA derives from its own short hairpin precursor molecule while a siRNA derives from long double-stranded RNA. Generally, siRNAs silence the same loci they derive from, whereas miRNAs silence heterogeneous genes [18]. miR34a is used as an inhibitor of non-small cell lung cancer (NSCLC) [19]. Moreover, RNAi is also of great importance in immunotherapy. For instance, MN-siPDL1 is known as a programmed death-ligand 1 (PD-L1) inhibitor, which downregulates PD-L1 to initiate cell apoptosis in the treatment of pancreatic cancer [20]. miR34a and miR29b also function as tumor suppressors in immunotherapy [21,22,23].

With the development of a non-viral gene delivery system, messenger RNAs (mRNAs) have become popular and promising agents in gene therapy. mRNAs are single-strand RNA molecules that regulate the synthesis of proteins in the cytoplasm via a process called translation. Precursor mRNAs (pre-mRNAs) are formed through transcription and then become mature mRNA via RNA splicing. During this process, introns are removed while exons are joined to form a contiguous coding sequence. Since the target of mRNA is the cytoplasm, the nuclear membrane is not a barrier for mRNA, which makes it advantageous compared to some other gene molecules, such as plasmid DNA [24]. mRNA has extensive applications in vaccine development [25,26,27,28,29,30], the innate immune response [24,31], and the treatment of cancers [32,33,34], virus infections [26], heart failure [35], and HIV [36,37]. In 2020, BNT162b2 and mRNA-1273, which are mRNA-based cationic lipid nanoparticles, have been approved for the COVID-19 vaccine. In addition, mRNA has been applied for gene editing. Remarkably, the progress of gene engineering technology has made more and more genomic disorders curable. Particularly, CRISPR (clustered regularly interspaced short palindromic repeats)-based genome editing, such as with CRISPR-Cas9 and CRISPR-Cpf1, contributes to a revolutionary breakthrough in this field [38,39,40,41,42,43,44]. Jennifer Doudna and Emmanuelle Charpentier won the Nobel Prize in chemistry in 2020 because of the discovery of the CRISPR-Cas9 gene technology tool [45]. Apart from those mentioned above, mRNA molecules still have breathtaking potentials in other therapeutic applications (Table 1). 

Though genetic agents have shown their clinical practice, delivering these agents as part of gene therapy faces many challenges. Apart from production costs, nucleic acids can be degraded easily in plasma by nucleases or other components, which reduces the efficiency of gene delivery [46]. As the first barrier, the extracellular matrix (ECM), including the interstitial matrix and basement membrane, plays a significant role in gene transport [47,48,49]. The deposition of ECM makes tumor tissues denser than normal tissues [50,51], which leads to an increase in interstitial tissue pressure and subsequently causes difficulties to gene transport [52,53]. Moreover, nucleic acids without any protections can be degraded by ribonucleases (RNases) in plasma [54,55]. Exogenous vesicles are also easily cleared by macrophages and dendritic cells [56,57,58,59,60,61]. Actually, it is mentionable that the mononuclear phagocytic system (MPS) plays an important role in the non-specific clearance of foreign nucleic acids [56,62]. MPS consists of macrophages, neutrophils, dendritic cells, blood monocytes, and granulocytes. It exists in many tissues, such as the liver, spleen, lymph nodes, and lung [63,64]. MPS can eliminate recognized exogenous materials from biological environments. Other extracellular barriers include excessive superfluous vasculature, vascular endothelial cells, and high interstitial fluid pressure [49,65,66]. Another challenge is the intracellular barrier, affecting mainly the cellular uptake, which is a process of interaction between extracellular materials and cell membrane. There are various pathways involved in cellular uptake, including macropinocytosis [67,68], clathrin-dependent endocytosis [69,70,71,72], caveolin-dependent endocytosis [73,74], and other clathrin/caveolin-independent endocytoses [75]. Once vesicles are endocytosed by the cells, they will be trapped by the endosome (Figure 1).

Lipid-based nanoparticles (LBNP) have been developed as a mature delivery platform for gene therapy during recent decades. They are tiny vesicles with a diameter range of 10–500 nm that is composed of biocompatible and biodegradable lipids. The suitable particle size enables LNP to escape mononuclear phagocyte system (MPS) uptake, subsequently prolonging the circulation time of LBNP and making LBNPs passively target tumor sites efficiently through an enhanced permeability and retention effect (EPR). Therefore, the off-target side effects and unnecessary harm to normal tissues can be minimized [76,77,78]. LBNPs can also promote endosomal escape after cellular uptake. Some LBNPs containing protonatable tertiary amine head groups with pKa values ranging from 6 to 7 can cause osmotically induced endosomal swelling and, subsequently, the burst of endosomes via the proton sponge effect [79,80,81]. LBNPs with highly fusogenic lipids can induce lipid fusion between the membranes of LBNPs and cells during structure phase transitions [3,79,82,83,84]. With the development of LBNP, it has become a mature and complicated drug delivery system that can encapsulate either small molecule components or nucleic acids, thus protecting molecules from clearance and overcoming intracellular and extracellular barriers, subsequently helping nucleic acids escape from the endosome and travel to target genes. For different purposes, various types of LBNP are designed, including for liposomes, lipid nanoparticles, solid lipid nanoparticles, lipid nanoemulsions, and nanostructured lipid carriers [85,86,87]. Apart from different kinds of natural or synthetic lipids, oils, wax, polymer conjugates, and steroids are also included. The advantages of these nano-sized formulations are not limited to enhancing drug stability, prolonging circulation, reducing toxicity, and controlling the release rate. They are also able to increase the local drug level at tumor sites, strengthen the drug bioavailability and selectivity, and decrease drug resistance [85].

Although the development of LBNP is promising, the pitfalls remain and need to be addressed. The low encapsulation efficiency of small molecules [88], cytotoxicity caused by cationic lipids [89,90], and systemic toxicity due to liver penetration are the primary causes that impair the further application of LBNP [91,92]. Due to its enhanced transfection efficiency and advantageous ability to go through extracellular and intracellular barriers as well as protect nucleic acids from degradation and elimination, LBNPs have still been explored and applied in gene delivery and gene editing. This review will introduce the influence of different components on the characteristics of LBNP, the modification strategies to overcome the existing pitfalls of LBNPs as nanoplatforms, and applications of LBNPs used for the treatment of different types of diseases, as well as some instances of gene therapy related to LBNP delivery system.

## 2. Modifications of Lipid-Based Nanoparticles

### 2.1. Types of Lipid-Based Nanoparticles

#### 2.1.1. Liposomes

Liposomes are spherical vesicles with the lipid bilayer, of which the external structure is like a cell, and are composed of multiple phospholipids. A phospholipid used to form liposomes has a hydrophilic head, which consists of a phosphate group, and two long hydrophobic tails consisting of a long hydrocarbon chain [93,94,95,96]. The characteristics of phospholipids mean that they can easily form a *w*/*o*/*w* lipid bilayer structure in an aqueous solution. Therefore, liposomes are able to encapsulate hydrophilic components stably in their internal aqueous phase, in which case those encapsulated components can hardly release from liposomes through the membrane. Some components with low aqueous solubility can also be entrapped in the aqueous core of the liposome by gradient-driven remote loading. The most famous instance is doxorubicin. In this strategy, there is an ammonium sulfate gradient between the internal phase of the liposome and the external solution, and the doxorubicin tends to go through the membrane and exchange ammonium with ammonium sulfate to form salt precipitation. Once the precipitation is formed inside the liposome, it will be stably trapped [97,98]. Doxil is the first drug loaded using the lipid-based nanoparticle that was approved by FDA in 1995. It shows great merits in minimizing the toxicity of doxorubicin when compared to the traditional formulation [99]. Another extensively applied drug is paclitaxel, and its clinical application was limited due to its low solubility and toxicity. Similarly, liposomal formulation encapsulating paclitaxel can enhance its aqueous solubility and reduce the toxicity, subsequently improving its clinical performance [100,101]. 

#### 2.1.2. Lipid Nanoparticles 

In comparison to the traditional liposomes, lipid nanoparticles (LNP) are advanced non-viral vectors used for the delivery of genetic medicines. They are sphere-shaped, nano-sized vesicles composed of one or more ionizable lipids. These lipids are positively charged, which enables the encapsulation of negatively charged nucleic acids in their internal aqueous phase [102,103,104]. The head of cationic lipids can be either quaternary amine or tertiary amine [105,106,107]. Cationic lipids containing quaternary amine are positively charged at both acidic pH and physiological pH, and it can trap nucleic acids such as messenger RNA (mRNA), small interfering RNA (siRNA), an antisense oligonucleotide (AS ON) to avoid the degradation of nucleic acids in vivo and enable endosomal escape after cellular uptake [87,103,108,109,110,111]. To reduce the cytotoxicity caused by ionizability, pH-sensitive cationic lipids containing tertiary amine are often used. These lipids are neutral at physiological pH, in which lipid nanoparticles will not cause cytotoxicity and process extracellular drug release, while they still enable endosomal escape in late endosome where the pH ranges from 5.5 to 6.5, so nucleic acids can target either the nucleus or cytosol [105,112,113,114,115,116]. LNPs have become a mature drug delivery system for gene therapy, with a series of advantages, such as high encapsulation efficiency, reduced toxicity, release control, enhanced cellular uptake, prolonged circulation time, and stability. Since 2012, the FDA has approved several ASOs carried by LNP for clinical use. In 2018, patisiran, the first mRNA drug for treating hereditary transthyretin-mediated amyloidosis, was encapsulated by cationic lipid nanoparticles and achieved significant success. It is a milestone for an RNAi-based lipid nanoparticle delivery system.

#### 2.1.3. Lipid Nanoemulsions

Lipid nanoemulsions (LNE) are colloidal droplet systems composed of oil, phospholipids, and an emulsifier, such as medium-chain triglycerides, which can reduce the interfacial tension on the droplet surface [117]. LNEs are normally used as carriers for lipophilic components. Like liposomes, LNEs are fully fledged drug delivery systems. However, the structure between LNE and liposome is different. An oil-in-water (O/W) LNE system consists of an interior oil core and a monolayer of phospholipids and emulsifiers. Drugs with poorly water solubility can dissolve in the interior liquid oil phase and be trapped, surrounded by a lipid monolayer. The hydrophilic head of phospholipids faces to exterior water phase so that LNEs can stably carry lipophilic drugs in an aqueous solution, thereby protecting the chemical stability of entrapped drugs [118,119,120,121]. Other types of lipid nanoemulsions, such as *w*/*o* or *w*/*o*/*w*, are also put into use for different purposes [121]. The compositions of LNE are biocompatible and biodegradable, so it will hardly induce toxicity or irritation [122]. LNEs can protect drugs from being trapped by the mononuclear phagocytic system, therefore improving the bioavailability of drugs [119,123,124]. The large surface area due to small particle size strengthens the absorption in vivo [118]. The lipophilic fusogenic properties promote cellular uptake as well as endosomal escape [125].

#### 2.1.4. Solid Lipid Nanoparticles

Solid lipid nanoparticles (SLN) are tiny spherical vesicles with a core, which can solubilize lipophilic components, surrounded by a layer of surfactants in the aqueous dispersion acting as stabilizers. To form a biological membrane, phospholipids are also utilized. The core of SLN, composed of glycerides, steroids, fatty acids, or waxes, remains solid at room temperature. The particle size of SLN can range from 50 to 1000 nm after encapsulation [126,127]. For the delivery of genetic drugs, cationic lipids are used to build a positive charge on the membrane surface. SLNs have a stronger ability to protect nucleic acids from degradation and leakage during storage compared with lipid nanoparticles [126,128,129]. They can encapsulate both hydrophilic and hydrophobic components because of their physical properties, which is helpful to lower the toxicity and improve the PK character [130].

#### 2.1.5. Nanostructured Lipid Carriers 

Nowadays, much research suggests that most of the encapsulated drugs may be attached or stranded on the outside layer surface instead of the solid core, in which the loading ability and stability of SLNs remain doubtful. As such, nanostructured lipid carriers are developed to improve drug loading efficiency and release characteristics based on SLNs [131,132,133]. NLCs are modified on the basis of SLNs and are different from SLNs because of the core, which is composed of both solid and liquid lipids at room temperature. Accordingly, NLCs can encapsulate components in both the solid phase and liquid phase simultaneously and subsequently control the drug release rate [134,135]. The above introduced lipid carriers are shown in (Figure 2). 

#### 2.1.6. Other Nanostructured Lipid Carriers 

Exosomes are nano-sized lipid vesicles secreted from living cells, ranging from 30 to 200 nm, present in cell culture medium and other biological fluids [136,137]. Exosomes are promising candidates for drug delivery platforms because of their high biocompatibility, blood–brain barrier (BBB) crossing capability, and low immunogenicity [138,139]. Exosomes carry various proteins and nucleic acids, reflecting their cell of origin; thus, the choice of donor cell type for exosomes is important if utilizing exosomes in drug delivery and diagnosis systems [140,141]. Avoiding immunomodulating activity and possible inflammation is one of the important criteria. Exosomes present in biofluids that contain other contaminations, thus requiring isolation, ultracentrifugation, sucrose density gradients, tangential flow filtration, size-exclusion chromatography, polymer-based precipitation, and immunoaffinity magnetic beads have been successfully used to isolate and purify exosomes [142,143]. Besides the natural cargos exosomes carry, engineering exosomes generated by cell transfection that contain enriched therapeutic biomaterials are interesting due to their potential high efficacy. Mainly, exosome-loaded nucleic acids (e.g., miRNA-155 and PTEN mRNA) have been widely used in cancer immunotherapy for tumor suppression by immune response regulation [144,145]. The summary of current drug delivery systems is listed in Table 2. 

### 2.2. Development and Modification Strategies

#### 2.2.1. Component Modification

For different purposes, LBNP can be designed using various strategies. Generally, one LBNP used for gene delivery usually consists of phospholipids, ionizable lipids, lipopolymers, and sterols.

Phospholipids are one of the most important components of the cell membrane, which consists of a hydrophilic phosphate head group, a glycerol bridge, and two hydrophobic fatty acid tails. Due to its amphiphilic characteristics, a lipid bilayer can be readily formed in the aqueous phase. On the basis of the modification of the head group, phospholipids can be divided into different subtypes, including phosphatidylcholine (PC), phosphatidylethanolamine (PE), phosphatidylserine (PS), phosphatidic acid (PA), phosphatidylinositol (PI), phosphatidylglycerol (PG), and cardiolipin (CL) [146]. Many researchers have reported that the saturation degree of lipids has an influence on the structure phase transition temperature (T_c_). The more unsaturated lipids have a lower phase transition temperature that makes lipids transform from a lamellar phase to an inverted hexagonal phase, which contributes to an enhanced fusogenicity [147,148,149]. The highly saturated lipid chain causes a high T_c_; for example, DSPC that contains two saturated fatty acid chains can form a cylindrical shape at room temperature, which enhances the stability and prolongs the circulation time. However, in some cases, it is hard to process drug release and endosomal escape due to less fusion with an anionic lipid membrane [150]. On the other hand, phospholipids such as 1,2-dioleoyl-sn-glycero-3-phosphocholine (DOPC) and 1,2-dioleoyl-sn-glycero-3-phosphoethanolamine (DOPE) with unsaturated fatty acid chains usually form inverted conical shapes at room temperature, and in this way the lipid nanoparticles containing these lipids are more likely to escape from the endosome. The alkyl chain length also has an effect on the physicochemical properties of LBNP. With the increase in the alkyl chain length, fewer drugs can be entrapped [151]. A long chain length can also enhance the stability of LBNP [152]. It is also mentioned that lipids with branched tails can strengthen the delivery efficiency of nucleic acids [153]. The head group is responsible for the pK_a_ value, the charge, and the amphiphilicity of phospholipids. Lipids with head groups such as phosphatidylcholine and phosphatidylethanolamine show a neutral charge, while lipids with phosphatidylserine and phosphatidylglycerol are negatively charged at physiological pH conditions [154]. The structure of head groups and alkyl tails determine the geometry of lipids, including phospholipids, cationic lipids, and ionizable lipids together. Lipids can form different structural phases in aqueous solution, such as hexagonal phases, lamellar phases, or inverted hexagonal phases, which can be divided by a packing parameter (P) decided by the ratio of hydrocarbon volume (V), the area of the head group (a), and the critical length of lipid tails (l_c_). The equation is P = V/a*l_c_ (Figure 3). Lipid-based nanoparticles containing lipids that adopt inverted hexagonal phases, such as DOPE, have better delivery efficiency since they can promote the lipid transition process from lamellar phase to non-lamellar phase and destabilize the structure of the endosomal membrane, which subsequently facilitates endosomal escape, and nucleic acids can be delivered to the cytosol [150,155,156]. 

In addition to phospholipids, cationic lipids are another important component used specifically for the delivery of nucleic acids due to their permanent positively charged head groups, which can encapsulate negatively charged nucleic acids in the internal aqueous phase through electrostatic self-assembly [157,158]. The cationic lipid nanoparticle carries nucleic acids through the cell membrane before being trapped in the endosome [159]. Then, the cationic lipids interact with the anionic phospholipids of the endosomal membrane to induce the disruption of the endosomal membrane and subsequently release the nucleic acids from the endosome. *N*-[1-(2,3-dioleyloxy)propyl]-*N*,*N*,*N*-trimethylammonium chloride (DOTMA) is the first cationic lipid used for gene delivery by Felgner et al. in 1987. They used liposomes consisting of DOTMA to encapsulate DNA. The DOTMA fuses with the negatively charged cell membrane and promotes endosomal escape [160]. *N*-[1-(2,3-dioleoyloxy)propyl]-*N*,*N*,*N*-trimethylammonium (DOTAP) is a quaternary ammonium lipid in which the linker is two ester bonds, instead of the ether bonds in DOTMA. In 1997, Templeton et al. applied liposome complexes containing DOTAP to improve DNA delivery [161]. Later, many studies have proven that DOTAP is a feasible and efficient cationic lipid in gene delivery [162,163,164,165]. Other cationic lipids apart from quaternary ammonium lipids have also been developed through the delocalization of cationic head groups, such as guanidinium lipids, imidazolium lipids, pyridinium lipids, piperidinium lipids, pyrrolidinium lipids, and phosphonium lipids [166,167,168,169,170,171,172]. 

Although the permanent positive charges bring cationic lipids high transfection efficiency and endosomal escape ability, they may cause severe cytotoxicity and short systemic circulation time [149]. To solve this problem, researchers proposed adjusting the charge ratio of lipid to nucleic acids [173,174,175]. However, the consequences of those attempts are not satisfying. To overcome the challenge, ionizable lipids have been developed for gene delivery. Unlike the cationic lipids with pH-independent positive charges, ionizable lipids normally are neutrally charged in a physiological pH (7.4) environment, while they become positively charged in an acidic pH environment. This is because the acid dissociation constant (pK_a_) of the tertiary amino head groups usually ranges from 6.0 to 7.0 [176]. They can electrostatically interact with nucleic acids in acidic aqueous solutions. Lipid nanoparticles containing pH-dependent ionizable lipids are neutral in plasma and normal tissues, and therefore they will hardly interact with anionic lipid membranes. The pH range in the endosome, as mentioned above, is 5.5 to 6.5, in which condition the ionizable lipids become positively charged and are able to interact with the endosomal membrane, helping nucleic acids escape from the endosome [110,177]. For example, 1,2-dioleyloxy-*N*,*N*-dimethyl-3-aminopropane (DODMA) is a pH sensitive ionizable lipid that has been well applied in the delivery of nucleic acids [113]. Hsu et al. used DODMA-based lipid nanoparticles and successfully delivered miRNA to a murine liver tumor [165]. Afterwards, 1,2-dilinoleyloxy-*N*,*N*-dimethyl-3-aminopropane (DLinDMA) and 1,2-dilinolenyloxy-*N*,*N*-dimethyl-3-aminopropane (DLenDMA) were developed to further improve the transfection efficiency. The difference between 1,2-distearyloxy-*N*,*N*-dimethyl-3-aminopropane (DSDMA), DODMA, DLinDMA, and DLenDMA is the saturation degree of their alkyl chains, with 0, 1, 2, and 3 double bonds on each chain, respectively [178]. Koynova R. et al. mentioned that LNPs containing DLinDMA or DLenDMA to deliver siRNA show better transfection efficiency than DODMA, while LNPs containing DSDMA show no gene silencing [179]. Apart from the saturation degree of the lipid tails, it is also remarked that lipids containing branched tails can promote endosomal escape better than those with linear tails [180,181]. Fenton et al. mentioned that longer unsaturated alkyl tails could enhance mRNA delivery efficiency. It is reported by Miao et al. that the incorporation of ester or alkyne groups rather than double bonds in the lipid tails can enhance the fusogenicity of the ionizable lipids [182]. The pKa value of ionizable lipids is also important for gene delivery efficiency. This can be determined through the modification of both head groups and hydrophobic tails. DLin-KC2-DMA, of which the pKa value is 6.7, is proven to be tenfold more potent than DLinDMA [150,183]. According to Jayaraman M. et al., the optimum pK_a_ value ranges from 6.2 to 6.5 [184]. The lipid nanoparticle used to encapsulate Onpattro contains DLin-MC3-DMA, which is one of the most effective ionizable lipids and is even more potent than KC2 [184,185]. DLin-MC3-DMA, of which the pK_a_ is 6.44, is shown to be the most active ionizable lipid among 56 amino lipid candidates [184]. Two COVID-19 vaccines named mRNA-1273 and BNT162b are encapsulated by LNPs containing SM-102 and Alc-0315, respectively (Figure 4). To balance the transfection efficiency and cytotoxicity, the researcher also utilizes both cationic lipids and ionizable lipids in one LNP system simultaneously. Yung et al. applied a combination of quaternary amine and tertiary amine cationic lipids to form lipid nanoparticles for the delivery of antimir-21, and it is successful for lung cancer treatment in vivo [105]. 

Sterols are a subtype of steroids and are important for stabilizing the structure of LBNP. Cholesterol is a representative that can fill the gaps between lipids to support the stable spherical structure, reduce the particle size of LBNP, and enhance the fluidity of the nanoparticle surface, and subsequently promote the fusion between LBNP and the cell membrane [111,186,187]. Nanoparticles containing cholesterol can also be taken up by the cell via low-density lipoprotein receptor (LDLR)-mediated endocytosis [188,189]. Zhigaltsev et al. used liposomes containing cholesterol to encapsulate doxorubicin and mentioned that high-ratio cholesterol triggers faster drug release [190]. Briuglia et al. also claimed that 2:1 is an optimum lipid–cholesterol ratio in order to achieve a suitable drug release rate [191]. Cholesterol can be modified via esterification and oxidation. It is reported that cholesterol variants, such as esterified cholesterol, can enhance RNA delivery efficiency [192,193]. Another cholesterol analogue used in lipid nanoparticles is β-sitosterol, which is a naturally occurring phytosterol that has similar structures to cholesterol, only with a modified C24 position [194]. LNP containing β-sitosterol has a polymorphic shape which enhances the fusogenicity of LNP, meaning that the liposomal membranes are more fluid than those containing cholesterol [194,195]. β-Sitosterol, as a composition in LNP, has been evaluated to have enhanced capability to escape from the endosome and deliver mRNA efficiently [196]. Other naturally occurring phytosterols include ergosterol, fucosterol, campesterol, and stigmastanol, all of which can be a composition of lipid nanoparticles for the improvement of RNA delivery performance [197,198]. 

#### 2.2.2. Surface Modification

LBNP surface modifications can increase the circulation time, enhance specific distribution and targeting ability, avoid particle aggregation, and promote cellular uptake. Lipopolymers are commonly used materials in nanoparticle surface modification that can increase the stability of nanoparticles in aqueous solutions [199,200]. Poly(ethylene glycol) (PEG) is a gold standard polymer used to avoid nanoparticle aggregation and increase the hydrophilic nature of nanoparticles in suspensions [201,202]. Nanoparticles coated with PEG are protected from phagocytosis and interaction with plasma proteins [203], and will not be recognized by MPS [204], which prolongs the circulation time and improves nanoparticle distribution [205]. PEG products have been investigated and widely used. Researchers usually bind PEG to an alkyl chain, of which the chain length and branches can be modified [3]. DSPE-PEG_2000_ was firstly used as one of the components of Doxil, which is the first FDA-approved liposomal drug in 1995 [206,207]. It was recently used as a component of Atu027 [208]. In 2020, Moderna applied DMG-PEG_2000_ in the lipid nanoparticle of the mRNA-1273 vaccine formulation [209,210]. To enhance the targeted drug delivery, Sun et al. conjugated PEG with folic acid in 2006 [211]. Folate receptors are reported to be overexpressed in cancer tissues [212,213], and nanoparticles conjugated with folic acid can be easily delivered [214]. PEG can be conjugated with proteins, which can specifically bind to transport receptors on the cell membrane to improve targeted delivery. For example, transferrin-conjugated PEG can improve cellular uptake by transferrin receptors [215,216]. However, PEG also faces several challenges, such as decreased drug activity and immunogenicity induction [217]. 

In addition to PEG, polyethylenimine (PEI) is another lipopolymer that has high transfection efficiency. The molecular weights and the amount of branches can be designed to possess different physical properties [218]. PEI can condense with nucleic acids through electrostatic self-assembly to form stable nano-sized particles [219]. PEI has a high buffering capacity and promotes cellular uptake and endosomal escape through the proton sponge effect. It functions like a sponge and pumps a large number of protons through the biological membrane, which, as a result, leads to an influx of chloride ions and water molecules and subsequently leads to an osmotic swelling and the bursting of the endosome [220,221]. Due to its properties, such as large molecule weight and high buffering capacity, PEI is not biocompatible enough and may also cause cytotoxicity [222,223]. To overcome the deficiencies of PEI, researchers conjugated PEG with PEI via covalent coupling to improve its performance as well [224,225,226]. Ligand modifications are also able to increase its transfection efficiency and biocompatibility. In 2016, Hu et al. modified PEI with trifunctional peptide R18 to enhance gene transfection efficiency [227]. Zheng et al. used transferrin-conjugated PEG-based LNP to deliver ASO LOR-2501 [228]. Conjugations with ionizable or neutral phospholipid cholesterols are frequently used as well [229,230,231]. 

Other surface modifications of lipids, including surface charge modification, targeting ligand modification, protein modification, and peptide modification strategies, are similar to surface modification of PEG and PEI mentioned above.

## 3. Applications of Lipid-Based Nanoparticles

### 3.1. LBNP in Treatment of Inherited Disease

The LBNP delivery system is a revolutionary breakthrough in gene therapy that can enable antisense oligonucleotides (ASOs), siRNA, mRNA, DNA, or gene-editing complexes, which may degrade easily in vivo, being delivered to their target to process anti-tumor activity via gene silencing, relevant protein expression, or genetic defect fixing [232]. Although LBNP can protect genetic drugs from degradation and elimination quickly, the main barrier that genetic drugs can hardly be delivered efficiently to target tissues still exists. Cytotoxicity is another issue that needs to be fixed before enabling clinical use. However, these barriers have been overcome recently. Antisense therapy has a long history. Since 1998, the first antisense oligonucleotide drug, called fomivirsen, which is an antiviral ASO that is used for the treatment of cytomegalovirus retinitis (CMV) in AIDS patients, was approved by the FDA. An important contribution to the development of ASOs has been made during the recent decades. To improve the therapeutic effect of ASOs and enhance their stability, various chemical modifications of ASOs have been processed for improved nuclease resistance and specifically target-binding affinity [233]. However, due to low solubility, weak cellular uptake, and unsatisfactory RNA–target affinity, the clinical use of ASOs was limited. Mipomersen is an apolipoprotein B-100 (ApoB-100) inhibitor used for the treatment of familial hypercholesterolemia (FH) [234]. FH is a genetic disease that is caused by mutations of apolipoprotein B [235]. This ASO can bind to mRNA coding for ApoB-100 and prevent the translation of ApoB-100 [236]. Afterward, other ASOs, such as golodirsen and volanesorsen, were approved by the FDA successively. However, the bioavailability of these ASOs without protection is not promising [237,238]. The most recent FDA-approved ASO drug is casimersen, which is used for the treatment of Duchenne muscular dystrophy (DMD) and works by inducing the exon 45 skipping of dystrophin so functional dystrophin can be translated [239,240,241].

The first FDA-approved application of LBNP for the RNAi-related therapy is patisiran, which is an LNP-based siRNA drug used for the treatment of polyneuropathy in people with hereditary transthyretin-mediated amyloidosis (ATTRm), which is a lethal disease caused by the mutation of transthyretin (TTR) gene expression [242]. Transthyretin is a transport protein found in serum and cerebrospinal fluid and predominantly synthesized in the liver [243,244]. TTR is normally a tetramer protein, which dissociates into monomers and leads to the formation of amyloid fibril and, subsequently, the accumulation of amyloid. Those insoluble fibrils aggregate at various tissues and finally result in ATTRm [244,245]. Patisiran is the first FDA-approved gene-silencing drug that can bind to the mRNA sequence of TTR and promote the degradation of target mRNA [246]. To target to mRNA specifically, patisiran is encapsulated into an LNP composed of DSPC (1,2-distearoyl-sn-glycero-3-phosphocholine), DLin-MC3-DMA (dilinoleylmethyl-4-dimethylaminobutyrate), mPEG_2000_-DMG (1,2-dimyristoyl-rac-glycero-3-methylpolyoxyethylene-2000), and cholesterol. This LNP formulation can avoid siRNA from degradation by RNase and help patisiran target hepatocytes. mPEG_2000_-DMG stabilizes the LNP structure and prevents the interaction with proteins and uptake via MPS clearance, subsequently prolonging the circulation time. DLin-MC3-DMA can enhance the fusogenicity of patisiran LNP. It can become positively charged in the endosome and interact with the endosomal membrane to facilitate siRNA to realize endosomal escape and to target the cytoplasm [164]. Afterward, other RNAi genetic drugs, such as givosiran, lumasiran, and inclisiran, have also been approved for clinical use. They are all siRNA drugs used for the treatment of acute hepatic porphyria, primary hyperoxaluria type 1 (PH1), and hypercholesterolemia, respectively [247,248,249,250]. Subsequently, as the second FDA-approved siRNA drug, givosiran, which targets aminolevulinate synthase 1 (ALAS1) mRNA, was used for the treatment of acute hepatic porphyria [251]. EPHARNA is a siRNA directed against ephrin type-A receptor 2. It is used for the treatment of advanced solid tumors. It is encapsulated by DOPC, which is a neutral lipid with 10 times greater efficiency than cationic liposomes [252].

Codon-optimized human frataxin (FXN) mRNAs were encapsulated in LNPs to treat Friedreich’s ataxia caused by the downregulation of FXN in 2016 [253]. In this study, adequate cellular uptake in hepatocytes and a remarkable expression of FXN protein was observed in adult mice after intravenous administration, and a recombinant human FXN protein was detected in the dorsal root ganglia 24 hours after 0.2 mg/kg intrathecal administration. Truong et al. developed LNPs encapsulating an mRNA encoding arginase 1 (ARG1) to treat inherited metabolic liver disorder arginase deficiency caused by biallelic mutations in ARG1 [254]. All the mice were ARG1-knockout mice and survived more than 11 weeks without hyperammonemia or body weight loss after continuous treatment of human codon-optimized ARG1 mRNA LNP, which indicates a potential therapeutic strategy for the treatment of arginase deficiency [254]. In 2019, another group used LNPs to deliver CYP7B1 mRNA as a treatment for hereditary spastic paraplegia type 5 (SPG5), a neurodegenerative disease caused by CYP7B1 gene loss-of-function mutations, and this resulted in noteworthy oxysterol degradation in liver and serum after two days of treatment, as well as the restoration of functional human CYP7B1 protein and the elimination of abnormal cholesterol metabolites [255]. These instances have proven that LBNPs serve as a good delivery system for the nucleic acid-based treatment of inherited diseases. 

### 3.2. LBNP in the Treatment of Infectious Diseases

mRNA vaccines have potent potentials to guard against infectious diseases caused by a virus-like influenza virus, Zika virus, rabies virus, human cytomegalovirus (CMV), and hepatitis C virus (HCV) [256]. They can cause significant and rapid immune response along with T cell induction. From previous liposome and solid lipid nanoparticles to ionizable lipid-based LNP, LBNPs have been developed and applied to be an advanced delivery system for RNA-based vaccines, with enhanced RNA loading capability and transfection efficiency. It has been well recognized that LBNP can protect mRNA from degradation and help mRNA go through the cell membrane via endocytosis and escape from the endosome by fusing with the endosomal membrane. 

Influenza is a viral infection that leads to damage to the respiratory system. An mRNA encoding influenza virus antigens encapsulated in LNPs can induce remarkable immune responses, including the expansions of central memory and effector memory CD4 and CD8 T cells against the influenza virus [257]. In 2013, Hekele et al. claimed that a self-amplifying mRNA (SAM) vaccine platform, which can encode the H1 hemagglutinin (HA) antigen from the H1N1 virus and the H7 HA antigen from the H7N9 virus, delivered by LNP could induce more rapid responses [258]. Mice receiving the first immunization showed considerable hemagglutinin inhibition and neutralizing antibody titers against the virus, and all mice had HI titers which are considered to be protective after the second immunization [258]. Afterward, Brazzoli et al. used cationic nanoemulsion to encapsulate SAM encoding influenza virus HA from H1N1 and induced strong immune responses, subsequently protecting mice immunized with this vaccine [259]. A modified non-replicating mRNA encoding influenza H10 was developed recently, which is encapsulated by LNP and can effectively induce an immune response [260]. It is also reported that circulating H10-specific memory B cells expanded after immunization [260]. SAMs encoding influenza antigens encapsulated by LNPs containing chitosan were delivered to DC and can also elicit the immune response after subcutaneous injection [261].

The rabies virus is a neurotropic virus that can cause irreversible damage to the central nervous system (CNS) in humans and animals. Due to the high occurrence of the rabies virus around the world, the development of nucleic acid-based vaccines has been driven over several years. Lutz et al. developed a sequence-optimized, chemically unmodified mRNA encoding rabies virus antigens encapsulated in LNPs that can induce protective antibody titers and remain stable for up to 1 year [262]. It is demonstrated that the innate immune response was activated remarkably at the injection site and in the draining lymph nodes (dLNs), and the functional antibody and T cell responses were stronger compared with the licensed vaccine [262]. Researchers designed a CV7201 and CV7202 (NCT03713086) mRNA-LNP formulation, consisting of cholesterol, DSPC, PEGylated lipid, and a cationic lipid, encoding the rabies virus glycoprotein (RABV-G) from a Pasteurized strain [263,264]. All participants showed measurable rabies neutralizing antibody responses after two 1 ug or 2 ug doses of CV7202 LNP [263].

The Zika virus (ZIKV) is a mosquito-borne arbovirus that emerged in 2015 as a severe pandemic around the world [265]. Pardi et al. designed an LNP-encapsulated nucleoside-modified mRNA encoding the pre-membrane and envelope glycoproteins derived from ZIKV that can induce potent and durable immune responses in mice and macaques after single low-dose intradermal immunization in 2017 [266]. Soon afterward, Richner et al. designed a modified mRNA vaccine LNP formulation encoding prM-E genes, which contributed to high neutralizing antibody titers that promote the immune response against ZIKV infection after single IM immunization in mice, while a second dose can elicit a more potent immune response [267]. In 2018, a replicating viral RNA encoding ZIKV antigens encapsulated by physico-chemically modified NLC was designed and induced a strong protective immune response after a single 10 ng IM dose in mice [268]. 

Since microbial infectious diseases show resistance to conventional antibiotics in many cases, LBNPs have been applied to improve the performance of antimicrobials. Vancomycin encapsulated by liposomes consisting of tetraether lipid can enhance the bioavailability of vancomycin 1 hour after oral administration on mice by about three fold, compared to free vancomycin [269]. Alhariri et al. successfully enhanced the efficacy of gentamicin by using anionic liposomes, which has better performance against planktonic P. aeruginosa and K. oxytoca, and which is better able to inhibit the formation of the biofilm of these strain than free gentamicin [270]. NLC has also been used to improve the biodistribution and affinity of rifampicin two fold against M. tuberculosis compared to free rifampicin [271].

LBNP can also deliver nucleic acid-based vaccines against other infectious diseases. Scientists used cationic nanoemulsion to deliver the human immunodeficiency virus (HIV) SAM vaccine encoding a clade C envelope glycoprotein, which induced remarkable cellular immune responses in rhesus macaques [272]. Cationic nanoemulsion containing DOTAP and squalene was also used to deliver nucleic acids encoding HCV, respiratory syncytial virus (RSV), and CMV, respectively [273]. The Ebola virus (EBOV) was firstly identified in 1976 and is a virus that can cause blood clots. During the past few years, many nucleic acid-based vaccine candidates against EBOV have been developed, one of which is an mRNA encoding EBOV glycoprotein (GP) encapsulated by LNP that induces EBOV-specific immunoglobulin G (IgG) and neutralizing antibody responses successfully [274]. Another vaccine candidate encapsulated by LNP against EBOV is siVP35-LNP, which can directly reduce viral load and promote an adaptive immune response to the virus [275]. Some nucleotide-modified mRNA-LNP vaccines (prME-mRNA, E80-mRNA, and NS1-mRNA) were also developed against the dengue virus-2 (DENV-2), and they successfully induce high levels of neutralizing antibodies and antigen-specific T cell responses [276]. 

### 3.3. LBNP in the Treatment of COVID-19

Severe acute respiratory syndrome coronavirus-2 (SARS-CoV-2) brought the coronavirus disease-19 (COVID-19) pandemic, which has caused 265 million infectious cases and over 5 million deaths. Consequently, the health care, business, and psychology of the entire world face big challenges. Since directly killing the virus is far from clinically possible, producing vaccines against SARS-CoV-2 is, thus, central in the race among the worldwide pharmaceutical companies and research institutes. mRNA vaccines are the most promising vaccines with the shortest manufacturing time, the most economical effectiveness, and clinically proven capability, and they have been developed to overcome the pandemic [277]. Both Moderna Biotechnology and Pfizer/BioNTech have developed mRNA vaccines in an unparalleled rapid manner and took less than one year to finish the manufacturing, efficacy testing, safety examination, and U.S. FDA Emergency Use Authorization approval [278]. 

The Moderna vaccine, mRNA-1273, demonstrated immunogenicity without safety concerns within trial limitations and a 94.1% efficacy for blocking COVID-19 among 30,420 clinical trial participants [279,280,281,282]. At the same time, BNT162b2, the Pfizer/BioNTech vaccine, has passed the clinical trial with 43,548 individuals with a 95% clinical efficacy in preventing COVID-19 illness, and the safety and anti-SARS-CoV-2 immune responses were also demonstrated [283,284]. Both mRNA vaccines were delivered using LNPs, and the compositions of LNPS are similar in that they both include an ionizable lipid, SM-102 (9-heptadecanyl 8 ((2 hydroxyethyl) (6 oxo 6—(undecyloxy) hexyl) amino) octanoate) for Moderna and ALC-0315 ((4-hydroxybutyl) azanediyl) bis (hexane-6,1-diyl) bis (2-hexyldecanoate) for Pfizer/BioNTech. The ionizable lipids are positive at the RNA complexation stage with a low pH because the lipids are tertiary amines that are protonated, and they are neutral at physiological pH, allowing efficient payload release and toxicity mitigation. Both vaccines have the 1,2-distearoyl-snglycero-3 phosphocholine (DSPC) as a helper lipid and cholesterol to facilitate packing cargos into the LNPs. PEG-DMG (1 monomethoxypolyethyleneglycol-2,3-dimyristylglycerol with polyethylene glycol) for Moderna and PEG-DMA (2 [(polyethylene glycol)-2000]-*N*,*N*-ditetradecylacetamide) for Pfizer/BioNTech are also applied to increase the circulation time by eliminating the clearance of LNPs by phagocytes and reducing the association of antibody and serum proteins [285]. Briefly, mRNAs in water and lipid mixtures in ethanol are mixed thoroughly in microfluidic devices. Subsequently, these two components form nanoparticles at pH 4.0 with an average size of 80 to 100 nm that encapsulates 100 negatively charged mRNA molecules per nanoparticle [110,186,286,287]. After two separate doses, intramuscularly administered, the host generates immune responses because of the presence of the SARS-CoV-2 spike glycoprotein that is encoded by the mRNA [281,288]. 

Compared with other vaccine platforms, mRNA vaccines have various advantages. First, the manufacturing is rapid, versatile, and flexible because it only requires the sequence encoding of the immunogen when generating different vaccines. Additionally, they are much safer compared with DNA-based vaccines because there is less possibility of mRNAs being integrated into the genome, since the translation happens in the cytoplasm, not the nucleus. In addition, the expression of target antigen after translation is much faster than other platforms [256,289,290]. Though promising progress has been made of mRNA vaccines in combating COVID-19, long-term effects are still unknown, and the long-term storage and delivery of mRNA vaccines are still an issue since mRNAs are not stable at room temperature [291]. 

### 3.4. LBNP in the Treatment of Cancer Immunotherapy 

In addition to inherited diseases, infectious diseases, and COVID-19, cancer immunotherapy is another area where LBNP-based therapies have been extensively applied in both preclinical and early clinical trials. Currently, LBNPs are often designed to carry multiple nucleic acids as cancer immunotherapeutic agents that target antigen-presenting cells (APCs) for the priming and activation of antigen-specific T cell-relative immune responses. Among a series of LBNPs, cationic LBNPs show great advantages in drug encapsulation and delivery when considering the intrinsic molecular charge of most involved nucleic acids. Chaudhuri et al. introduced lysinylated cationic amphiphiles covalently grafted with mannose-mimicking shikimoyl and quinoyl groups in the head group region for stable and efficient DNA loading and long-term immune responses [292]. In another study, Sayour’s group developed an RNA–liposomal (RNA-NPs) cancer vaccine with personalized tumor-derived mRNA (representing a tumor-specific transcriptome) encapsulated in DOTAP nanoparticles [293]. Despite the great merits of efficient encapsulation, the surface charge of cationic LBNP has impaired their further use. This is because nanoparticles with a positive charge tend to affect lymphatic transport and permeation through tissues after injection, as they will immobilize within the negatively charged extracellular matrix, thus inducing hemolysis and platelet aggregation [294]. Accordingly, many studies began to localize cationic LBNP in vivo. For example, Kranz et al. significantly decreased the cumulative charge of mRNA and lipid-based nanoparticles (composed of DOTMA and DOPE) by regulating the ratio of lipid:mRNA [295]. According to their results, near-neutral mRNA–liposomes systems with slightly positive charging are able to localize within the spleen, instead of aggregating in regions such as the heart and lung. The interaction between loaded mRNA and different LBNP delivery platforms was systematically evaluated to determine the best combination for the efficient uptake and expression of the encoded antigen by DC populations. Consequently, the mRNA incorporated in this study can activate Toll-like receptor 7 (TLR7) on DCs to induce the secretion of IFN and desired anti-cancer responses. An initial phase I trial based on this study demonstrated that IFNα and strong antigen-specific T cell responses could be induced at low-dose levels in patients with advanced malignant melanoma (NCT02410733) (Table 3).

These promising pre-clinical and early clinical outcomes suggest the effective regulation of the physicochemical properties of LBNPs and optimized interaction with delivered cargos. Like previous applications we discussed, challenges in LBNPs-based cancer immunotherapy also include uncontrollable release profiles, which could lead to low bioavailability and efficiency of the delivery system. As such, corresponding solutions to solve the uncontrollable releasing behaviors and associated mechanisms are necessary for better LBNPs for cancer immunotherapy in future clinical stages.

## 4. Conclusions and Future Perspectives

In recent decades, the importance of LBNP in gene delivery has been realized, and multiple strategies to improve these vehicles have been applied. In this review, we have discussed multiple kinds of LBNPs in various applications, particularly gene therapy and immunotherapy, with the roles of LBNP highlighted. Because of the significant advantages of gene delivery, research has been carried out for the purpose of improving vesicle stability, cellular uptake, and endosomal escape. In recent years, modification strategies rendered LBNP a mature and outstanding gene delivery system with feasible production procedures and acceptable costs. With the wide use of LBNP, not only gene therapies, but also conventional therapies, such as chemotherapy and immunotherapy to treat cancers, inherited diseases, infections, and immunodeficiencies, have been advanced as well. Despite the success in many preclinical studies and clinical cases, many barriers remain. The main challenge is that the ratio of contents such as helper lipids, ionizable lipids, and cholesterol that can process endosomal escape and be delivered to the target is far from being enough. The therapeutic effect could be further enhanced if this barrier is overcome. Additionally, although LBNP can protect drugs from degradation and excretion, the accumulation of the LBNP in the desired sites is limited, which leads to undesirable therapeutic effects and potential side effects. Moreover, there is a safety issue; for instance, some components in LBNP may cause irritation and inflammation and activate host immune responses. In addition, cationic lipids can cause cytotoxicity, and the short circulation time of the delivered LBNPs; these are other challenges that need much attention. In these cases, more fundamental studies are required. Once the understanding of LBNP in gene delivery is strengthened, the LBNP-based drug delivery platform will become more promising and advance in future clinical applications. 

## Figures and Tables

**Figure 1 molecules-27-01943-f001:**
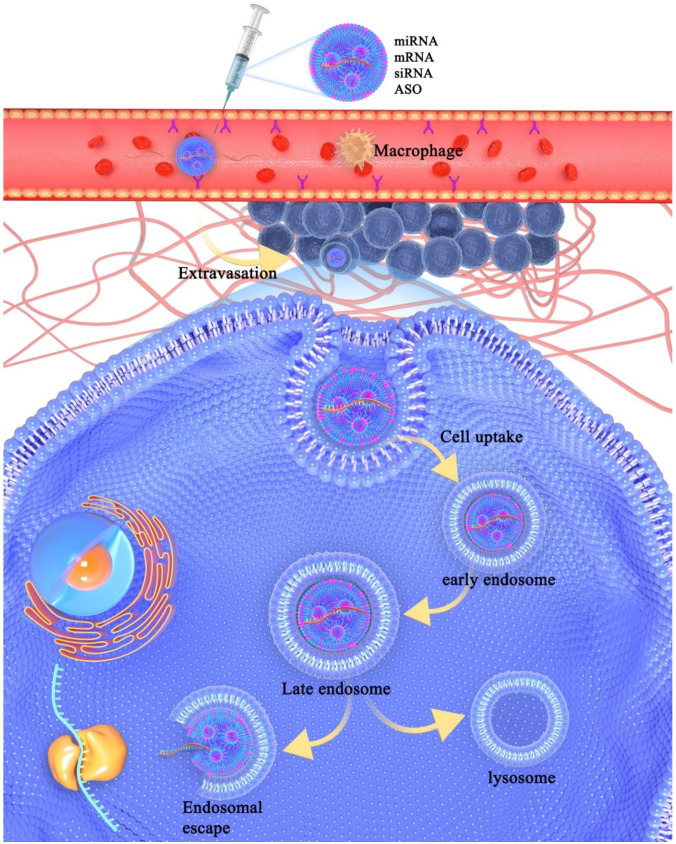
Schematic demonstration of extracellular and intracellular barriers during the process of nucleic acids delivery to the cytosol. Once nucleic acids are injected into the vein, the LBNP will protect them from being degraded by RNase in plasma and eliminated by macrophage. LBNP can carry nucleic acids go through the cell membrane via receptor-mediated endocytosis. The components such as cationic lipids and ionizable lipids can fuse with endosomal membrane and help nucleic acids release from endosome and deliver to target site.

**Figure 2 molecules-27-01943-f002:**
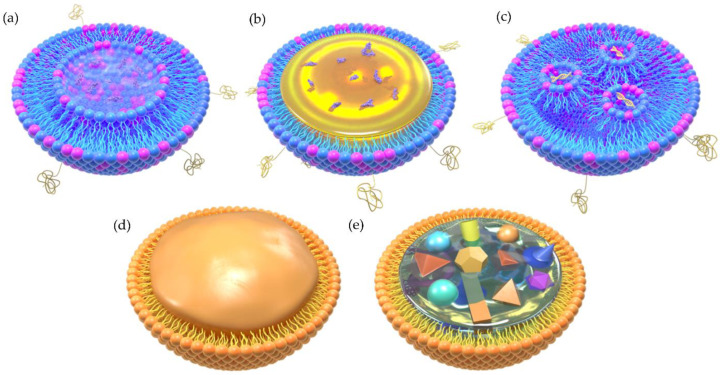
Schematic illumination of the two-dimensional structure of different types of LBNPs. (**a**) Liposome, (**b**) lipid nanoemulsion (LNE), (**c**) lipid nanoparticle (LNP), (**d**) solid lipid nanoparticle (SLN), (**e**) nanostructured lipid carrier (NLC).

**Figure 3 molecules-27-01943-f003:**
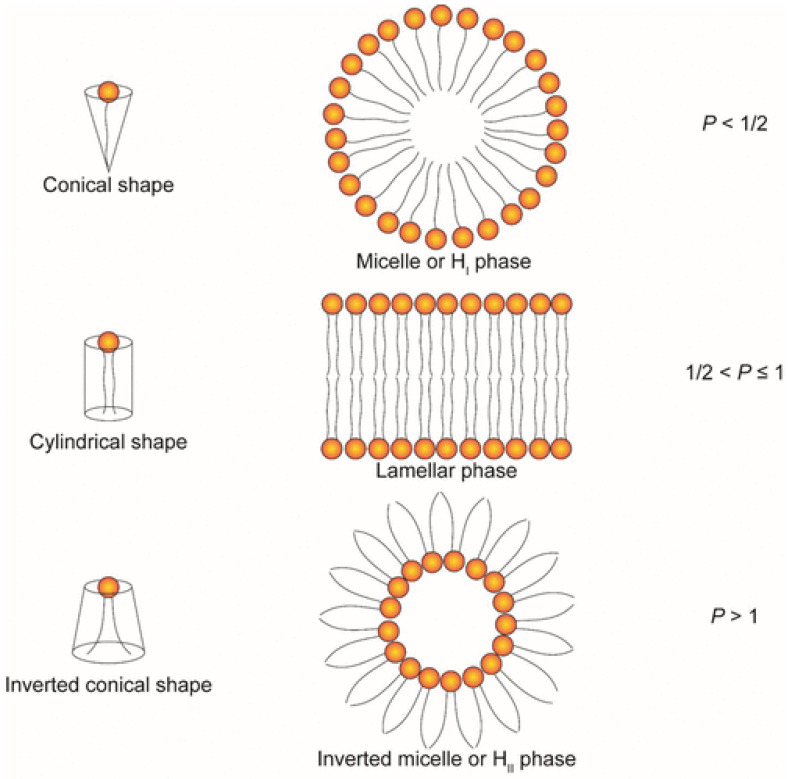
The schematic demonstrates different structural phases under different conditions [3]. *p* = v/a*l_c_. If *p* < ½, lipids with conical shape are more likely to adopt hexagonal phase. If ½ < *p* < 1, lipids with cylindrical shape tend to adopt a lamellar phase. If *p* > 1, inverted conical-shaped lipids will adopt an inverted hexagonal phase [3,155]. Copyright 2022 American Chemical Society.

**Figure 4 molecules-27-01943-f004:**
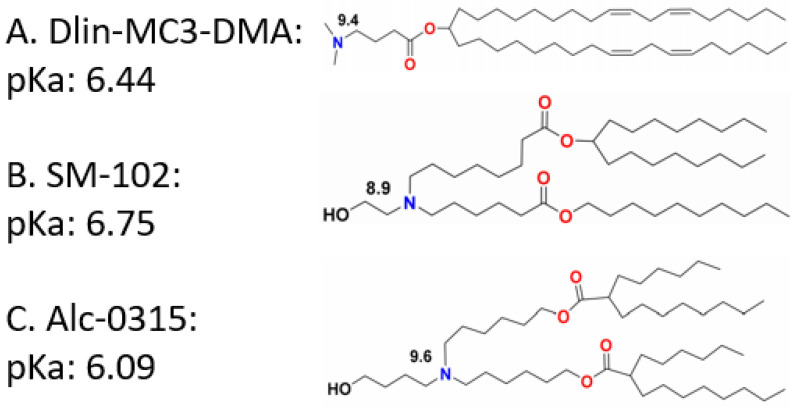
Structure and pK_a_ value of DLin-MC3-DMA (**A**), SM-102 (**B**), and Alc-0315 (**C**). Copyright is reserved by 1996–2022 MDPI (Basel, Switzerland).

**Table 1 molecules-27-01943-t001:** Representative of FDA-Approved RNA Agents for Clinical Use.

Genetic Drugs	Target	Indication	Clinical Trials Identifier or Approved Year
siRNA			
Patisiran (ALN-TTR02)	Transthyretin	Hereditary transthyretin-mediated amyloidosis	FDA-approved in 2018
Givosiran (ALN-AS1)	5-Aminolevulinic acid synthase	Acute hepatic porphyria	FDA-approved in 2019
Lumasiran (ALN-GO1)	HAO1	Primary hyperoxaluria type 1	FDA-approved in 2020
Inclisiran (ALN-PCSSC)	PCSK9	Hypercholesterolemia	FDA-approved in 2021
ALN-VSP02	VEGF-A, KSP	Solid tumors	NCT01158079
ARB-001467	HBsAg	Hepatitis B, chronic	NCT02631096
TKM-PLK1	PLK1	Adrenocortical carcinoma, hepatocellular carcinoma	NCT01437007
siRNA-EphA2-DOPC	EphA2	Advanced or recurrent solid tumors	NCT01591356
Atu027	Protein kinase N3	Advanced solid cancer	NCT00938574
ND-L02-s0201	HSP47	Hepatic fibrosis	NCT03241264
DCR-MYC	Oncogene MYC	Solid tumors, hepatocellular carcinoma	NCT02314052
PRO-040201	Apo-B	Hypercholesterolemia	NCT00927459
mRNA			
mRNA-1273	SARS-CoV-2	COVID-19 vaccine	FDA-approved in 2022
BNT162b2	SARS-CoV-2	COVID-19 vaccine	FDA-approved in 2021
mRNA-2416	OX40L	Metastatic solid tumor, lymphoma	NCT03323398
mRNA-2752	OX40L	Lymphoma, ovarian	NCT03739931
mRNA-1647	6 CMV	Cytomegalovirus infection	NCT03382405
Antisense oligonucleotides			
Mipomersen	ApoB-100	Homozygous familial hypercholesterolemia	FDA-approved in 2013
Nusinersen	SMN2 gene	Spinal muscular atrophy	FDA-approved in 2016
Golodirsen	Dystrophin	Duchenne muscular dystrophy	FDA-approved in 2019
Volanesorsen	Apo-CIII	Familial chylomicronaemia syndrome	NCT02658175
Viltolarsen	Exon 45	Duchenne muscular dystrophy	FDA-approved in 2020
Casimersen	Exon 45	Duchenne muscular dystrophy	FDA-approved in 2021

**Table 2 molecules-27-01943-t002:** Current Drug Delivery Systems.

Drug Delivery System	Characteristics	Application
Liposome	Spherical vesicles with lipid bilayer	Delivery of small molecule drug, oligo nucleotide
Lipid nanoparticle	Multilamellar and faceted nano-sized vesicle with *w*/*o*/*w* phase	Delivery of nucleic acids for gene therapy, gene editing, and genomic engineering
Lipid nanoemulsion	Spherical vesicles with a lipid monolayer and a lipid core	Delivery of hydrophobic or unstable drugs
Solid lipid nanoparticle	Spherical vesicles with a core consisting of solid matrix	Delivery of Hydrophobic drugs, active lipid ingredients, and nucleic acids
Nanostructured lipid carrier	Spherical vesicles with a core consisting of both solid and liquid lipids	Delivery of virus, non-viral nucleic acids, small molecule drugs with low aqueous solubility
Exosome	Nano-sized lipid vesicles secreted from living cells, ranging from 30 to 200 nm, present in cell culture medium and other biological fluids	Carrier of various proteins and nucleic acids for intercellular communication

**Table 3 molecules-27-01943-t003:** FDA-Approved Drugs Encapsulated by Lipid-Based Nanoparticles.

Products	Formulation	FDA Approval Year	Mechanism	Indication
Doxil	Liposomal doxorubicin	1995	Topoisomerase II inhibitor	Leukemias, multiple myeloma, Hodgkin’s lymphoma, various cancers
DaunoXome	Liposomal daunoribucin	1996	Topoisomerase II inhibitor	Various cancers, HIV-associated Kaposi’s sarcoma
AmBisome	liposomal amphotericin B	1997	Binding to ergosterol and cause ion leakage	Invasive fungal infection
Marqibo	Liposomal vincristine	2012	Tubulin inhibitor	Lymphoma, leukemia, melanoma, brain cancer
ONIVYDE	Liposomal irinotecan	2015	Topoisomerase I inhibitor	Colon cancer, small-cell lung cancer
Lipusu	Liposomal paclitaxel	2016	Microtubule inhibitor	Breast cancer, non-small-cell lung cancer
Vyxeos	Liposomal daunorubicin and cytarabine	2017	Topoisomerase II inhibitor, antimetabolic	Acute myeloid leukemia (AML)
ONPATTRO	Patisiran siRNA LNP	2018	siRNA target to transthyretin	Hereditary transthyretin-mediated amyloidosis
GIVLAARI	Givosiran siRNA LNP	2019	siRNA target to 5-aminolevulinic acid synthase	Acute hepatic porphyria
OXLUMO	Lumasiran siRNA LNP	2020	siRNA target to HAO1	Primary hyperoxaluria type 1
Leqvio	Inclisiran siRNA LNP	2021	siRNA target to PCSK9	Hypercholesterolemia
mRNA-1273	mRNA LNP	2022	Encoding SARS-CoV-2 S protein	COVID-19 vaccine
BNT162b2	mRNA LNP	2021	Encoding SARS-CoV-2 S protein	COVID-19 vaccine

## Data Availability

No new data was generated.

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
