# Peer review of "Modification of Lipid-Based Nanoparticles: An Efficient Delivery System for Nucleic Acid-Based Immunotherapy"

_molecules, 2022, doi:10.3390/molecules27061943_

Round 1
Reviewer 1 Report
In this manuscript, the authors summarized the characteristics and modification of lipid-based nanoparticle, and their application in gene therapy, which has also been mentioned in other previous publications. The authors should include the discussion of these works and reflect what the present work has different from the ones that have already appeared previously. Accordingly, I would like to recommend a minor revision before the manuscript can be accepted.
Point 1: I recommend the authors to give more discussion on the clinical application of current nucleic acids delivery concerning lipid-based nanoparticles.
Point 2: The authors should also mention cancer immunotherapy as one of the application of gene therapy involving lipid-based nanoparticle.
Point 3: I recommend the authors add more summary in the end of the introduction part, explaining the advantages and pitfalls of lipid-based nanoparticles.
Point 4: The authors mentioned that lipid-based nanoparticles may cause some safety issues, which should be discussed a little more like what kind of safety issues can be caused.
Point 5: I recommend the authors to explain the illustration pictures with more details, especially the first picture.
Point 6: Some spell and grammar errors need to be corrected.
Author Response
Reviewer #1
Point 1: I recommend the authors to give more discussion on the clinical application of current nucleic acids delivery concerning lipid-based nanoparticles.
A: Thank you for your advice. The current clinical application of gene delivery with LBNP has been supplemented in Table 1 of the revised manuscript.
Point 2: The authors should also mention cancer immunotherapy as one of the application of gene therapy involving lipid-based nanoparticle.
A: Thank you for your advice. We added one paragraph (session 3.4) introducing the application of gene therapy involving LBNP in the field of cancer immunotherapy.
Point 3: I recommend the authors add more summary in the end of the introduction part, explaining the advantages and pitfalls of lipid-based nanoparticles.
A: Thank you for the suggestion. We further discussed the advantages of LBNP based on the previous version.
“Due to its enhanced transfection efficiency and advantageous ability to go through ex-tracellular and intracellular barriers as well as protect nucleic acids from degradation and elimination, LBNPs have still been explored and applied in gene delivery and gene editing.”
Point 4: The authors mentioned that lipid-based nanoparticles may cause some safety issues, which should be discussed a little more like what kind of safety issues can be caused.
A: Thank you for the advice. We added some discussion in the conclusion part associated with the side effects probably caused by LBNPs.
“Moreover, the safety issue, for instance, some components in LBNP may cause irrita-tion and inflammation and activate host immune responses. In addition, cationic lipids can cause cytotoxicity, and short circulation time of the delivered LBNPs are other challenges that need much attention.”
Point 5: I recommend the authors explain the illustration pictures with more details, especially the first picture.
A: Thanks for the advice. We have added some explanations to the captions of figure 1 and figure 2.
“Once nucleic acids are injected into the vein, the LBNP will protect them from being degraded by RNase in plasma and eliminated by macrophage. LBNP can carry nucleic acids go through the cell membrane via receptor-mediated endocytosis. The components such as cationic lipids and ionizable lipids can fuse with endosomal membrane and help nucleic acids release from the endosome and deliver to the target site.”
Point 6: Some spell and grammar errors need to be corrected.
A: Sorry for the mistakes we made. We have carefully revised the manuscript and corrected language issues.

Reviewer 2 Report
In general, this manuscript covers an inteersting topic suitable for publication in molecules. However, there are some aspects taht must be adressed before its publication:
- Figure 1 should be explained with more details. Please, include this information in Figure 1 caption.
- If table 1 shows FDA approved RNAs, I suggest removing NCT number. If they are approved for that indication, they are approved. Maybe, the authors can include approved year. What about RNA approved by EMA? It could be also interesting show this information.
- Lipomes encapsulating antimicrobial agents are approved. They should be mentioned.
- Figure 2 must be explained with more details, showing the type of carrier that represent ecah image.
- Quality of figure 3 is low. It must be improved.
- A table showing lipid nanocarriers approved by FDA and EMA should be included.
- A table showing all the systems discussed in this manuscript should also be included.
Author Response
Reviewer #2
Point 1: Figure 1 should be explained with more details. Please, include this information in Figure 1 caption.
A: Thank you for the suggestion. We have added some explanations to the captions of figure 1 and figure 2
“Once nucleic acids are injected into the vein, the LBNP will protect them from being degraded by RNase in plasma and eliminated by macrophage. LBNP can carry nucleic acids go through the cell membrane via receptor-mediated endocytosis. The components such as cationic lipids and ionizable lipids can fuse with endosomal membrane and help nucleic acids release from the endosome and deliver to the target site.”
Point 2: If table 1 shows FDA approved RNAs, I suggest removing NCT number. If they are approved for that indication, they are approved. Maybe, the authors can include approved year. What about RNA approved by EMA? It could be also interesting show this information.
A: Thank you for the suggestion. We have already deleted the NCT number of those FDA-approved products, with the approved year of these drugs supplemented in the revised manuscript.
Point 3: Liposomes encapsulating antimicrobial agents are approved. They should be mentioned.
A: Thank you for the advice. We have supplemented these applications.
“Since microbial causing infectious disease show resistance to conventional antibiotics in many cases, LBNPs have been applied to improve the performance of antimicrobials. Vancomycin encapsulated by liposomes consisting of tetraether lipid can enhance the bioavailability of vancomycin 1 hours after oral administration on mice about 3-fold, compared to free vancomycin. () Alhariri et al. successfully enhanced the efficacy of gentamicin by using anionic liposomes, which has better performance against planktonic P. aeruginosa and K. oxytoca, as well as to inhibit the formation of biofilm of these strain compared to free gentamicin. (10.2147/IJN.S141709) NLC has also been used to improve the biodistribution and affinity of rifampicin 2-fold against M.tuberculosis than free rifampicin. (10.1016/j.colsurfb.2018.12.003)
Point 4: Figure 2 must be explained with more details, showing the type of carrier that represents each image.
A: Sorry for the negligence. We have labeled number beside each type of carrier and described the basic characteristics below Figure 2.
Point 5: The quality of figure 3 is low. It must be improved.
A: Sorry for the mistake. Figure 3 was originally cited from another review written by Luc Wasungu in 2006(doi: 10.1016/j.jconrel.2006.06.024). We cited another figure from another review written by Yuebao Zhang et al., of which the quality should be improved.
Point 6: A table showing lipid nanocarriers approved by FDA and EMA should be included.
A: Thank you for the suggestion. We have added Table 3 in the revised manuscript.
Table 3 FDA-approved Drugs Encapsulated by Lipid-Based Nanoparticle
|
Products |
Formulation |
FDA-approved year |
Mechanism |
Indication |
|
|
Doxil |
Liposomal doxorubicin |
1995 |
Topoisomerase II inhibitor |
Leukemias, Multiple myeloma, Hodgkin’s lymphoma, Various cancers |
|
|
DaunoXome |
Liposomal daunoribucin |
1996 |
Topoisomerase II inhibitor |
Various cancers, HIV-associated Kaposi’s sarcoma |
|
|
AmBisome |
liposomal Amphotericin B |
1997 |
Binding to ergosterol and cause ion leakage |
Invasive fungal infection |
|
|
Marqibo |
Liposomal vincristine |
2012 |
Tubulin inhibitor |
Lymphoma, leukemia, melanoma, brain cancer |
|
|
ONIVYDE |
Liposomal irinotecan |
2015 |
Topoisomerase I inhibitor |
Colon cancer, small cell lung cancer |
|
|
Lipusu |
Liposomal paclitaxel |
2016 |
Microtubule inhibitor |
Breast cancer, non-small-cell lung cancer |
|
|
Vyxeos |
Liposomal daunorubicin and cytarabine |
2017 |
Topoisomerase II inhibitor, antimetabolic |
Acute myeloid leukemia (AML) |
|
|
ONPATTRO |
Patisiran siRNA LNP |
2018 |
siRNA target to transthyretin |
Hereditary transthyretin-mediated amyloidosis |
|
|
GIVLAARI |
Givosiran siRNA LNP |
2019 |
siRNA target to 5-aminolevulinic acid synthase |
Acute hepatic porphyria |
|
|
OXLUMO |
Lumasiran siRNA LNP |
2020 |
siRNA target to HAO1 |
Primary Hyperoxaluria type1 |
|
|
Leqvio |
Inclisiran siRNA LNP |
2021 |
siRNA target to PCSK9 |
Hypercholesterolemia |
|
|
mRNA-1273 |
mRNA LNP |
2022 |
encoding SARS-COV-2 S protein |
COVID19 vaccine |
|
|
BNT162b2 |
mRNA LNP |
2021 |
encoding SARS-COV-2 S protein |
COVID19 vaccine |
|
Point 7: A table showing all the systems discussed in this manuscript should also be included.
A: Thanks for the advice. We have added Table 2, which summarizes all the drug delivery systems mentioned in this review.
Table 2 Current Drug Delivery System
|
Drug delivery system |
Characteristics |
Application |
|
Liposome |
Spherical vesicles with lipid bilayer |
Delivery of small molecule drug, oligonucleotide |
|
Lipid nanoparticle |
Multilamellar and faceted nano-sized vesicle with W/O/W phase 10.1021/acs.nanolett.0c01386 |
Delivery of nucleic acids for gene therapy, gene editing and genomic engineering |
|
Lipid nanoemulsion |
Spherical vesicles with a lipid monolayer and a lipid core |
Delivery of hydrophobic or unstable drugs |
|
Solid lipid nanoparticle |
Spherical vesicles with a core consisting of the solid matrix |
Delivery of Hydrophobic drugs, active lipid ingredients, and nucleic acids |
|
Nanostructured lipid carrier |
Spherical vesicles with a core consisting of both solid and liquid lipids 10.3109/21691401.2014.909822 |
Delivery of virus, non-viral nucleic acids, small molecule drugs with low aqueous solubility |
|
Exosome |
Nano-sized lipid vesicles secreted from living cells, ranging from 30 to 200 nm, present in cell culture medium and other biological fluids |
Carrier of various proteins and nucleic acids for intercellular communication |

Reviewer 3 Report
Title: Modification of lipid-based nanoparticle: an efficient delivery system for nucleic acids-based immunotherapy
The review written by Chi Zhang et al summarized the recent advances and remaining challenges in the development of Lipid-based nanoparticles (LBNPs) for nucleic acids-based immunotherapy. First, the authors have given concise introduction about gene therapy and the strategies of gene therapy (delivering genetic agents like ASOs, siRNA, mRNA, miRNA and plasmid-DNA into cells and target to nucleus or cytosol). Later, the authors focused on modifications of lipid-based nanoparticles including types of lipid-based nanoparticles (for example: Liposomes, Lipid nanoparticles, Lipid nanoemulsions, Solid lipid nanoparticles and Nanostructured lipid carriers) and development of modification strategies (e.g., component modification and surface modification). Finally, the authors discussed about application of lipid-based nanoparticles (in treatment of inherited diseases, infection diseases and COVID-19). This review can be published in “Molecules”. If possible, I suggest the authors to give figures for each type of strategy and LBNPs applications, so that readers can understand easily. Overall, the review is well written and all the references are in proper format.

Author Response
The review written by Chi Zhang et al summarized the recent advances and remaining challenges in the development of Lipid-based nanoparticles (LBNPs) for nucleic acids-based immunotherapy. First, the authors have given concise introduction about gene therapy and the strategies of gene therapy (delivering genetic agents like ASOs, siRNA, mRNA, miRNA and plasmid-DNA into cells and target to nucleus or cytosol). Later, the authors focused on modifications of lipid-based nanoparticles including types of lipid-based nanoparticles (for example: Liposomes, Lipid nanoparticles, Lipid nanoemulsions, Solid lipid nanoparticles and Nanostructured lipid carriers) and development of modification strategies (e.g., component modification and surface modification). Finally, the authors discussed about application of lipid-based nanoparticles (in treatment of inherited diseases, infection diseases and COVID-19). This review can be published in “Molecules”. If possible, I suggest the authors to give figures for each type of strategy and LBNPs applications, so that readers can understand easily. Overall, the review is well written and all the references are in proper format.
R: Thank you for considering the manuscript as the publication for the molecules journal. We have checked the grammer in the revised the manuscript.